# Factors affecting guardians' decision making on clinic-based purchase of children's spectacles in Nigeria

**Ving Fai Chan**[1,2]*, **Ai Chee Yong**[1], **Ciaran O'Neill**[1], **Christine Graham**[1], **Nathan Congdon**[1,3,4], **Lynne Lohfeld**[1,5], **Tai Stephan**[4], **Anne Effiom Ebri**[6]

1 Centre for Public Health, School of Medicine, Dentistry and Biomedical Sciences, Queen's University Belfast, Northern Ireland, United Kingdom, 2 College of Health Sciences, University KwaZulu Natal, Durban, South Africa, 3 Zhongshan Ophthalmic Centre, Sun Yat-sen University, Guangzhou, China, 4 Orbis International, New York, New York, United States of America, 5 Clinical and Epidemiological Eye Research Center, Wenzhou Medical University, Wenzhou, China, 6 Brien Holden Vision Institute Foundation (Africa) Trust, Durban, South Africa

* v.chan@qub.ac.uk

## Abstract

### Background

This study aims to understand the key factors influencing guardians' decisions when purchasing spectacles for their children in semi-urban and urban areas of Cross River State, Nigeria, where a spectacle cross-subsidisation scheme will be implemented.

### Methods

This cross-sectional study was conducted among all consecutive guardians visiting the Calabar (urban), Ugep, Ikom and Ogoja (semi-urban) public eye clinics in Cross River State, southern Nigeria, from August 1 to October 31 2019, and whose children had significant refractive errors (myopia ≤-0.50D, hyperopia ≥1.50D, astigmatism >0.75D) and received spectacles. Guardians were interviewed using a questionnaire which included i) close-ended questions on reasons guardians choose to purchase spectacles for their children in eye clinics, ii) guardians' perceptions of the quality and design of children's current spectacle, iii) factors most heavily influencing their choice of spectacles for children, and iv) open-ended questions to seek guardians' suggestions on how to improve the current spectacle range.

### Results

All 137 eligible guardians (67.2% women [n = 92]) who visited the selected eye clinics participated in the study (response rate = 100%), with 109 (79.6%) from semi-urban and 28 (20.4%) attending urban clinics. Guardians from both urban and semi-urban clinics prioritised frame design, quality, and material as the main factors affecting their decision when purchasing spectacles for their children. Female guardians and those with higher incomes were both 1.5 times more likely to emphasise frame quality when describing selection

**Data Availability Statement:** The data underlying the results presented in the study are available from Queen's University Belfast Data Repository

(DOI: 10.17034/6f84899d-fafb-40dd-becd-998abea781f9).

**Funding:** Initials of the authors who received each award - VFC, AEE, NC, CON Grant numbers awarded to all authors - R3138CPH The full name of funder - Department for the Economy, Global Challenge Research Council (United Kingdom) URL- https://www.qub.ac.uk/Research/Our-research/InternationalResearch/ The funders had no role in study design, data collection and analysis, decision to publish, or preparation of the manuscript.

**Competing interests:** VFC is a Trustee of Vision Aid Overseas, a non-governmental organisation involved in delivering refractive services in LMICs, including Africa. NC is the Director of Research for Orbis International, an organisation involved in delivering eyecare, including children's refractive services, in Africa and other settings. The conflict of interest stated in the manuscript does not alter our adherence to PLOS ONE policies on sharing data and materials. We have included this in both the manuscript and covering letter.

criteria for purchasing spectacles for their children than male guardians (p = 0.01) or guardians earning less (p = 0.03).

## Conclusion

Design, material, and frame quality are key factors influencing guardians when purchasing spectacles for their children in these setting and female guardians or those with higher income prioritise frame quality. This study could guide the planning and implementation of a novel cross-subsidisation scheme in Cross River State.

## Introduction

Despite spectacles being a highly cost-effective and safe intervention to manage refractive error (RE) [1], 61% of global vision loss is due to uncorrected refractive error (URE) [2]. Poor vision caused by URE negatively impacts physical mobility, educational attainment, and psychosocial well-being while creating financial burdens for families, communities, and society [3–5]. Limited access to affordable refractive services is widely recognised as a barrier to spectacle uptake, particularly in low- and middle-income countries (LMICs) [6–8].

The estimated prevalence of refractive error in Nigeria, where the present study was conducted, is between 5 and 8%, and URE remains the most common cause of visual impairment nationally among school-going children [9–11]. The main barriers to the uptake of refractive services following a child's failing vision screening in that country are the cost of eye care services, logistical issues such as transport, and long waiting times at clinics [12]. To address the challenge of URE, the Comprehensive Child Eye Health in Nigeria (CCEHiN) programme was implemented in 11 of Nigeria's 36 states, including Cross River State, from 2017 to 2020. The programme included eye health promotion, school-based vision screening, and referral for free spectacles at designated town-level Child Eye Clinics situated in Calabar, Ugep, Ogoja, and Ikom in Cross River State.

As a continuation of the CCEHiN programme, the Cross River State government aims to implement a scheme at public hospitals to subsidise the cost of inexpensive spectacles for low-income families, using the profits derived from selling more expensive frames. Understanding guardians' decision-making criteria is essential for programme implementers to procure a range of spectacles meeting local needs. Limited studies on this topic suggest that parents in rural China prioritise quality and design of spectacles [13], while in South Africa, design, staff recommendations, and quality are the main factors driving decision-making [14].

To assist with the design of the children's spectacles cross-subsidisation strategy that will be implemented in CRS, a four-part study was conducted. It collected collect data on child eye morbidities prevalence; barriers to follow-up eye examinations and spectacle uptake; parents' willingness to pay for their children's spectacles; and the current study which aims to understand the key factors influencing guardians' decisions when purchasing spectacles for their children in semi-urban (Ikom, Ugep and Ogoja) and urban (Calabar) areas. These four clinics serve the rural, semi-urban and urban population in CRS. We also assessed guardians' perceptions of service quality at four public child eye clinics, one per community, to facilitate improvements.

## Materials and methods

The study was approved by Medical Research and Ethics Committees at Queen's University Belfast (Pre FREC Ref 19.24v3) and by the Cross River State Ministry of Health's Health Research and Ethics Committee (CRS/MH/HREH/019/Vol.V1/175). We employed a convenient finite sampling where all guardians visiting participating eye clinics from 1 August to 31 October 2019 were interviewed if their children had significant REs (myopia $\leq$-0.50D, hyperopia $\geq$1.50D, astigmatism >0.75D) and received spectacles. Guardians of children who did not need a pair of spectacles were excluded. Written informed consent was obtained from the guardians, and interviews were conducted in meeting rooms at the eye clinics.

Four interviewers and four optometrists underwent training by VFC and AEE to ensure consistency in data collection. To ensure guardians could appropriately contextualise the potential benefit of spectacles for their children, they were informed at the beginning of the interview that (i) their child's reduced vision was caused by RE, (ii) according to the optometrist, spectacle correction was the appropriate treatment to improve their child's vision, and (iii) their responses to the survey would not be known to anyone other than the interviewer, affect the quality of spectacles they could choose or their receiving spectacles at no cost.

Guardians' demographic information was collected and included sex, age, occupation, educational level, monthly household income, status as a wage earner, guardian's history of and satisfaction with spectacle wear, and guardian's history of an eye examination. Children's information such as sex, age, presenting vision, and the total number of siblings was also collected.

To ensure standardisation, study personnel followed a prepared script when orally delivering the survey to guardians. The survey was conducted using a questionnaire (S1 File) adapted from Pillay et al.'s[14] study in South Africa. The questionnaire was first reviewed by a panel of experts from the field of public health, ophthalmology, anthropology and optometry, and then piloted with ten local community members to ensure its appropriateness, relevance, comprehensiveness. Minor changes were made to include closed-ended questions on why guardians chose to purchase particular spectacles for their children in eye clinics and their perceptions of the quality and design of children's current spectacle (5-point Likert-scale, 1 = Very bad and 5 = excellent) in the questionnaire. Also included were questions on factors most heavily influencing their choice of spectacles for children regarding design, quality, material, staff recommendation, price, and brand of frames. Open-ended questions asked guardians for suggestions on how to improve each of these six areas.

Statistical Package for the Social Sciences V25 (SPSS Inc., Chicago, IL) was used for data management and analysis. Guardian's age and monthly household income, child's age and presenting vision, and the number of siblings were categorised and coded. Each guardian was asked an open-ended question on their actual monthly income. After discussing a suitable cutoff for low vs high income with local partners, we then regrouped them into two categories: $\leq$ Nigerian Naira 100,000 (USD240), and > Nigerian Naira 100,000. Differences in demographic characteristics between guardians in varying socioeconomic settings were compared using the t-test for continuous variables and chi-square test for categorical variables. Feedback on existing services provided at the clinics and factors affecting decision-making were presented using frequency distribution (percentages). Associations between guardians' demographic characteristics and purchasing decisions were assessed using logistic regressions. Qualitative responses regarding the factors most heavily influencing guardians' selection of children's spectacles were grouped into main themes, and the frequencies of responses for each theme were tabulated. A p-value of $< 0.05$ was considered significant throughout.

## Results

All 137 eligible guardians (mean age 42.5 ± 7.70 years, 67.2% women [n = 92]) who visited the selected eye clinics participated in the study (response rate = 100%), with 109 of them (79.6%) from semi-urban child eye clinics and 28 (20.4%) from urban settings. Guardians from urban clinics were significantly more likely to have a tertiary-level education (p = 0.006), work as civil servants (p = 0.043) and earn more than USD240 per month (p<0.001) compared to those recruited at semi-urban clinics. Demographic characteristics such as sex, age, status as a wage earner, and history of spectacle wear did not differ significantly between guardians from the different settings (Table 1).

The majority of semi-urban guardians (n = 67, 42.4%) decided to purchase their spectacles at the children's eye clinics because they were convenient for them as a comprehensive one-stop service. In contrast, urban guardians listed other reasons (n = 30, 58.8%), such as availability of less expensive treatments, trust in the quality of care, friendly staff attitude, and having been referred to the clinics from children's schools. However, more than half (n = 17, 60.7%) of guardians seen at urban clinics reported that they were not informed about available frame choices, compared to only 14.2% at semi-urban clinics who did not receive this information.

More than 95% of guardians in both groups rated the design and quality of available frames as good or excellent, though they suggested that the materials used should be more flexible and lighter (35.5%), and that a wider selection of children's frame sizes and colours should be offered (40.9%). Nearly all guardians (99.2%) indicated they would return to the clinic to purchase another pair of spectacles, and all of them (100%) reported they would recommend the clinic to their friends and families (Table 2).

In semi-urban areas, guardians reported that they made their purchase decisions based on the design (28.3%), quality (20.8%), and material (17.3%) of frames. Guardians from urban areas were most concerned about frame quality (32.1%), followed by material (27.4%) and design (23.8%). Price, staff recommendations, and brand were least influential in decision-making in both settings (Table 3).

Female guardians and those with higher incomes were both 1.5 times more likely to place more emphasis on frame quality than male guardians (p = 0.01) or guardians earning less (p = 0.03). Guardians of female children were twice as likely to prioritise price compared to guardians of male children (p = 0.01). Guardians with more children reported that they were more heavily reliant on staff recommendations than guardians with fewer children (p = 0.006) (Table 4).

## Discussion

Our study's primary aim was to understand the factors affecting guardians' decision-making when purchasing spectacles for their children in Cross River State, southern Nigeria. Results indicate that both urban and semi-urban guardians prioritise frame design, quality, and material as the top three factors affecting their decision when choosing spectacles for their children. These findings are consistent with reports from China [13] and South Africa [14], although parents in rural China prioritise good spectacle quality most highly and price least often [13], whereas, in South Africa, frame design and quality are the most important factors [14].

These findings have practical implications. Historically, patients' desire for lower prices outweighed the importance of style and design when choosing spectacles. Growing evidence shows that price is no longer the most influential factor affecting spectacle wear compliance. Rather, evidence from studies in China [13], India [15] and Mexico [16], reveals that unattractive frames are frequently not worn by children prescribed spectacles.

**Table 1. Demography of participants from four eye clinics (n = 137).**

| | CLINICS n (%) | | | Test of significance, p-value |
|---|---|---|---|---|
| | **SEMI-URBAN** | **URBAN** | **Total** | |
| | **n = 109 (79.6%)** | **n = 28 (20.4%)** | **n = 137 (100%)** | |
| **Guardians' sex** | | | | |
| Female | 69 (63.3%) | 23 (82.1%) | 92 (67.2%) | Pearson Chi-square, p = 0.058 |
| Male | 40 (36.7%) | 5 (17.9%) | 45 (32.8%) | |
| **Guardians' age (years)** | | | | |
| 40 years and younger | 40 (36.7%) | 15 (53.6%) | 55 (40.1%) | Pearson Chi-square, p = 0.104 |
| Older than 40 years | 69 (63.3%) | 13 (46.4%) | 82 (59.9%) | |
| **Mean ± SD** | 43.00 ± 7.7 | 40.50 ± 7.6 | 42.49 ± 7.7 | T-test, p = 0.126 |
| **Guardians' highest School attainment** | | | | |
| No formal schooling | 7 (6.4%) | - | 7 (5.1%) | Fisher's Exact Test, p = 0.006* |
| Primary and secondary school | 48 (44.0%) | 5 (17.9%) | 53 (38.7%) | |
| College/university | 54 (49.5%) | 23 (82.1%) | 77 (56.2%) | |
| **Guardians' occupation** | | | | |
| Civil servant | 21 (19.3%) | 10 (35.7%) | 31 (22.6%) | Fisher's Exact Test, p = 0.043* |
| Skill artisan | 23 (21.1%) | 1 (3.6%) | 24 (17.5%) | |
| Teacher | 28 (25.7%) | 7 (25.0%) | 35 (25.5%) | |
| Trader | 33 (30.3%) | 8 (28.6%) | 41 (29.9%) | |
| Unemployed | 3 (2.8%) | - | 3 (2.2%) | |
| Others | 1 (0.9%) | 2 (7.1%) | 3 (2.2%) | |
| **Wage earner** | | | | |
| Yes | 93 (85.3%) | 27 (96.4%) | 120 (87.6%) | Fisher's Exact Test, p = 0.330* |
| No | 14 (12.8%) | 1 (3.6%) | 15 (10.9%) | |
| No response | 2 (1.8%) | - | 2 (1.5%) | |
| **Guardians' gross monthly income** | | | | |
| ≤ Nigerian Naira 100,000 (USD240) | 79 (72.5%) | 15 (53.6%) | 94 (68.6%) | Fisher's Exact Test, p<0.001* |
| > Nigerian Naira 100,000 (USD240) | 15 (13.8%) | 13 (46.4%) | 28 (20.4%) | |
| No response | 15 (13.8%) | - | 15 (10.9%) | |
| **Median monthly income (IQR) ‡** | 50,000 (20,000–80,000) | 90,000 (30,000–192,500) | 50,000 (27,500–100,000) | Mann-Whitney U, p = 0.055# |
| **Guardians' glasses wear history** | | | | |
| Never | 56 (51.4%) | 15 (53.6%) | 71 (51.8%) | Fisher's Exact Test, p = 1.000* |
| Yes | 52 (47.7%) | 13 (46.4%) | 65 (47.4%) | |
| No response | 1 (0.9%) | - | 1 (0.7%) | |
| **Guardians' last eye examination** | | | | |
| Never had an eye exam before | 49 (45.0%) | 10 (35.7%) | 59 (43.1%) | Pearson Chi-square Test, p = 0.564 |
| Less than six months | 32 (29.4%) | 11 (39.3%) | 43 (31.4%) | |
| More than six months | 28 (25.7%) | 7 (25.0%) | 35 (25.5%) | |
| **Child's age (years)** | | | | |
| 12 years and younger | 74 (68.5%) | 19 (67.9%) | 93 (68.4%) | Pearson Chi-square Test, p = 0.947 |
| Older than 12 years | 34 (31.5%) | 9 (32.1%) | 43 (31.6%) | |
| **Child's sex** | | | | |
| Female | 65 (59.6%) | 12 (42.9%) | 77 (56.2%) | Pearson Chi-square Test, p = 0.111 |
| Male | 44 (40.4%) | 16 (57.1%) | 60 (43.8%) | |
| **Child's Presenting VA (LogMAR) before correction** | | | | |
| 0.30 or better in the better eye | 77 (70.6%) | 15 (53.6%) | 92 (67.2%) | Pearson Chi-square Test, p = 0.086 |
| 0.31 or worse in the better eye | 32 (29.4%) | 13 (46.4%) | 45 (32.8%) | |

*(Continued)*

**Table 1.** (Continued)

| | CLINICS n (%) | | | Test of significance, p-value |
|---|---|---|---|---|
| | **SEMI-URBAN** | **URBAN** | **Total** | |
| | **n = 109 (79.6%)** | **n = 28 (20.4%)** | **n = 137 (100%)** | |
| **Child's number of sibling/s** | | | | |
| 0–3 | 94 (86.2%) | 24 (85.7%) | 118 (86.1%) | Fisher's Exact Test, p = 0.206* |
| 4–6 | 13 (11.9%) | 2 (7.1%) | 15 (10.9%) | |
| ≥ 7 | 2 (1.8%) | 2 (7.1%) | 4 (2.9%) | |

The majority of guardians surveyed in the current study are satisfied with the existing frame quality and designs offered at the child eye clinics and reported they would return for a second pair of spectacles and recommend the clinics to relatives and friends. These findings are extremely important as they show that the current frame selection can meet local needs. It is also well-understood that consumers' satisfaction with a product is highly associated with consumer loyalty [17].

Considering the socioeconomic differences in our service area, the proposed cross-subsidisation strategy will provide guardians with frames in the high-, medium-, and low-priced range. Despite their having higher income and being more educated, urban guardians are motivated by the lower prices of spectacles in the government programme's clinics. In a survey at health facilities in urban Nigeria, 42% of 214 adult spectacle wearers agreed that available spectacles were expensive and identified price as a challenge to them purchasing glasses [18]. This fits with Holden et al.'s observation that spectacles in LMICs are often not readily affordable to most of the local population [19]. In the current setting, considering that spectacles provided at eye clinics are cheaper than those offered in private shops, it seems likely that the lower price will be attractive to guardians. However, findings show that price is not the only consideration when guardians select spectacles for their children. The proposed cross-subsidisation strategy may create competition for these private facilities, driving prices down even further.

Good service is also vital to ensure continued patronage by guardians such as those in the current study. A one-stop comprehensive facility offering refractive and optical services is highly favoured by many guardians, especially in semi-urban clinics. Our findings corroborate those of Pillay et al. [14], whereby in settings where optical services are limited and geographical distance is a common barrier to access, as for rural dwellers in Nigeria [20], our proposed cross-subsidisation spectacle scheme is potentially very attractive to guardians. However, this programme must be supported by appropriately trained staff to ensure that guardians are informed of available frame choices. Results from this study highlight the current lack of such information on choice.

Our study's strengths include capturing the views of both semi-urban and urban guardians, providing data on purchasing behaviours across different socioeconomic levels. Secondly, by using a mix of closed- and open-ended questions, we obtained several practical suggestions for future improvements of frame quality and design and customer service. The study captured consumers' preferences on frame features, such as lighter frame materials and a wider range of frame sizes and colours.

Limitations to our study must also be acknowledged. Firstly, our study's small sample size may not detect more modest associations among variables [21]. Secondly, some guardians might have known that free spectacles were offered at the clinics, leading them not to rank price as important a factor as they might otherwise for fear of having to pay for spectacles later.

**Table 2. Participants' feedback on the existing services provided at the child eye clinics.**

| | CLINICS | | |
|---|---|---|---|
| | **SEMI-URBAN** | **URBAN** | **Total** |
| | **n (%)** | **n (%)** | **n (%)** |
| **Reason for getting glasses here:** | | | |
| It was convenient | 67 (42.4) | 9 (17.6) | 76 (36.4) |
| I like the spectacle frames | 13 (8.2) | - | 13 (6.2) |
| The service was good | 39 (24.7) | 9 (17.6) | 48 (23.0) |
| I didn't know I could get it elsewhere | 13 (8.2) | 3 (6.0) | 16 (7.7) |
| Others | 26 (16.5) | 30 (58.8) | 56 (26.7) |
| **Total responses[‡]** | 158 (100.0) | 51 (100.0) | 209 (100.0) |
| **Informed of spectacle frames** | | | |
| Yes | 91 (85.8) | 11 (39.3) | 102 (76.1) |
| No | 15 (14.2) | 17 (60.7) | 32 (23.9) |
| **Total responses** | 106 (100.0) | 28 (100.0) | 134 (100.0) |
| **Opinion on frame design** | | | |
| Excellent | 24 (22.6) | 9 (33.3) | 33 (24.8) |
| Good | 81 (76.5) | 17 (63.0) | 98 (73.7) |
| Neutral | 1 (0.9) | 1 (3.7) | 2 (1.5) |
| Bad | - | - | - |
| Very bad | - | - | - |
| **Total responses** | 106 (100.0) | 27 (100.0) | 133 (100.0) |
| **Opinion on frame quality** | | | |
| Excellent | 23 (21.7) | 7 (25.9) | 30 (22.6) |
| Good | 78 (73.6) | 19 (70.4) | 97 (72.9) |
| Neutral | 5 (4.7) | 1 (3.7) | 6 (4.5) |
| Bad | - | - | - |
| Very bad | - | - | - |
| **Total responses** | 106 (100.0) | 27 (100.0) | 133 (100.0) |
| **Improvement on existing frames** | | | |
| Brand choice | 7 (10.1) | 1 (4.2) | 8 (8.6) |
| Design | 12 (17.5) | 2 (8.3) | 14 (15.1) |
| Material [ɥ] | 25 (36.2) | 8 (33.3) | 33 (35.5) |
| Others [ꬔ] | 25 (36.2) | 13 (54.2) | 38 (40.9) |
| **Total responses[‡]** | 69 (100.0) | 24 (100.0) | 93 (100.0) |
| **Return in future** | | | |
| Yes | 104 (99.0) | 27 (100.0) | 131 (99.2) |
| No | 1 (1.0) | - | 1 (0.8) |
| **Total responses** | 105 (100.0) | 27 (100.0) | 132 (100.0) |
| **Recommend the clinics** | | | |
| Yes | 104 (100.0) | 26 (100.0) | 130 (100.0) |
| No | - | - | - |
| **Total responses** | 104 (100.0) | 26 (100.0) | 130 (100.0) |

[‡] Each participant can give more than one reason/option.

[ꬔ] Reasons: because of free treatment, because this is a hospital, referred, general health check, government establishment offers a better price, the staff is friendly.

[ꬔ] Others: casing for frames, improve on all, satisfied with the stock, frame sizes, quality, durable.

[ɥ] Material: lighter, flexible.

**Table 3. Factors influencing decision making in the sample (total responses = 367).**

| Factors influencing decision making | Semi-urban | Urban | Total |
|---|---|---|---|
| | n (%) | n (%) | n (%) |
| Design | 80 (28.3) | 20 (23.8) | 100 (27.2) |
| Material | 49 (17.3) | 23 (27.4) | 72 (19.6) |
| Quality | 59 (20.8) | 27 (32.1) | 86 (23.4) |
| Staff recommendation | 43 (15.2) | 5 (6.0) | 48 (13.1) |
| Price | 37 (13.1) | 3 (3.6) | 40 (10.9) |
| Brand | 15 (5.3) | 6 (7.1) | 21 (5.7) |
| Total | 283 | 84 | 367 |

**Table 4. Relationship between demography profiles and factors influencing guardians' spectacles purchasing decision.**

| Demographic profiles | Factors influencing guardians' decision | | | | | |
|---|---|---|---|---|---|---|
| | Design Odd ratio | Material Odd ratio | Quality Odd ratio | Price Odd ratio | Staff recommendation Odd ratio | Brand Odd ratio |
| Location | | | | | | |
| Semi-urban | 1 | 1 | 1 | 1 | 1 | 1 |
| Urban | 0.85 | 1.59 | 1.52 | 0.31 | 0.4 | 1.4 |
| *p-value* | p = 0.662 | p = 0.001 | p<0.001 | p = 0.013 | p = 0.085 | P = 0.374 |
| Guardian's sex | | | | | | |
| Female | 1 | 1 | 1 | 1 | 1 | 1 |
| Male | 1.2 | 1 | 0.69 | 1 | 0.76 | 0.67 |
| *p-value* | p = 0.096 | p = 0.969 | p = 0.011 | p = 0.947 | p = 0.285 | p = 0.363 |
| Guardian's age | | | | | | |
| 40 years and younger | 1 | 1 | 1 | 1 | 1 | 1 |
| Older than 40 years old | 1.15 | 0.88 | 0.93 | 1.03 | 0.71 | 0.5 |
| *p-value* | p = 0.219 | p = 0.389 | p = 0.533 | p = 0.873 | p = 0.126 | p = 0.093 |
| Guardians' highest school attainment | | | | | | |
| No schooling | 1 | 1 | 1 | - | 1 | 0 |
| Primary or secondary school | 0.8 | 3.64 | 2.1 | 1 | 1.09 | 0.59 |
| College or University | 0.91 | 4.21 | 2.41 | 0.661 | 0.7 | 1 |
| *p-value* | p = 0.469 | p = 0.068 | p = 0.077 | p = 0.236 | p = 0.155 | p = 0.275 |
| Guardians' occupation | | | | | | |
| Civil servant | 1 | 1 | 1 | 1 | 1 | 1 |
| Skill artisan | 1.06 | 0.65 | 0.45 | 0.64 | 1.53 | 1.05 |
| Teacher | 1 | 0.77 | 0.44 | 0.53 | 0.58 | 0.15 |
| Trader | 1.15 | 0.91 | 0.77 | 0.58 | 1 | 1.05 |
| Unemployed | 0.47 | 0.51 | 1.06 | 0.73 | 0.87 | 1.65 |
| Others | 1.41 | 0.51 | 0.71 | - | 0.87 | 0 |
| *p-value* | p = 0.399 | p = 0.482 | p<0.001 | p = 0.352 | p = 0.118 | p = 0.121 |
| Wage earner | | | | | | |
| No | 1 | 1 | 1 | 1 | 1 | 1 |
| Yes | 1.28 | 1.73 | 0.79 | 1.6 | 0.9 | 0.56 |
| *p-value* | p = 0.198 | p = 0.05 | p = 0.092 | p = 0.593 | p = 0.211 | p- = 0.482 |
| Guardians' gross monthly income | | | | | | |
| ≤ Nigerian Naira 100,000 (USD 263) | 1 | 1 | 1 | 1 | 1 | 1 |

*(Continued)*

**Table 4.** (Continued)

| Demographic profiles | Factors influencing guardians' decision | | | | | |
|---|---|---|---|---|---|---|
| | Design Odd ratio | Material Odd ratio | Quality Odd ratio | Price Odd ratio | Staff recommendation Odd ratio | Brand Odd ratio |
| > Nigerian Naira 100,000 (USD 263) | 1.09 | 1.22 | 1.47 | 0.82 | 1.03 | 1.36 |
| *p-value* | p = 0.125 | p = 0.387 | p = 0.031 | p = 0.248 | p = 0.973 | p = 0.715 |
| Guardians' worn glasses before | | | | | | |
| Yes | 1 | 1 | 1 | 1 | 1 | 1 |
| Never | 1.13 | 0.83 | 1.03 | 1.54 | 0.74 | 1.45 |
| *p-value* | p = 0.502 | p = 0.295 | p = 1.000 | p = 0.193 | p = 0.249 | p = 0.464 |
| Guardians' last eye examination | | | | | | |
| Never had an eye exam before | 1 | 1 | 1 | 1 | 1 | 1 |
| Less than six months ago | 1.03 | 0.66 | 1 | 1.04 | 0.33 | 0.53 |
| More than six months ago | 1.17 | 1.14 | 1.1 | 1.92 | 1.09 | 0.9 |
| *p-value* | p = 0.424 | p = 0.037 | p = 0.82 | p = 0.058 | p = 0.003 | p = 0.473 |
| Child's age | | | | | | |
| ≤ 13 years old | 1 | 1 | 1 | 1 | 1 | 1 |
| > 13 years old | 1.1 | 0.89 | 1 | 0.54 | 1.14 | 1.36 |
| *p-value* | p = 0.458 | p = 0.516 | p = 0.951 | p = 0.06 | p = 0.617 | p = 0.501 |
| Child's sex | | | | | | |
| Female | 1 | 1 | 1 | 1 | 1 | 1 |
| Male | 1.13 | 1 | 1.05 | 0.49 | 0.76 | 1.22 |
| *p-value* | p = 0.431 | p = 0.422 | p = 0.681 | p = 0.012 | p = 0.383 | p = 0.377 |
| Child's Presenting VA | | | | | | |
| Less than 0.31 LogMAR | 1 | 1 | 1 | 1 | 1 | 1 |
| 0.31 LogMAR or worse in the better eye | 0.89 | 0.93 | 1.31 | 0.84 | 1.53 | 1 |
| *p-value* | p = 0.278 | p = 0.665 | p = 0.051 | p = 0.567 | p = 0.07 | p = 0.979 |
| Child's number of sibling/s | | | | | | |
| 0–3 sibling/s | 1 | 1 | 1 | 1 | 1 | 1 |
| 4–6 siblings | 0.79 | 1.02 | 0.94 | 0.67 | 2.09 | 0.81 |
| ≥ 7 siblings | 1.32 | 1.92 | 1.17 | 1.67 | 2.34 | 1.56 |
| *p-value* | p = 0.214 | p = 0.208 | p = 0.91 | p = 0.459 | p = 0.006 | p = 0.728 |

To minimise the potential for uninformed or biased responses, trained interviewers explained the study objectives and relevant background information before starting interviews, following a standard script. Thirdly, we recognized that some parents may not have brought their children for treatment at one of the local hospitals and so their views may have been unrepresented. We addressed this concern in another study in which we interviewed guardians who did not bring their children for a follow-up eye examination at a hospital eye clinic. The adults were asked about reasons for not seeking additional treatment and what would encourage them to do so [22].

## Conclusion

The current study highlights that placing one-stop comprehensive services at public child eye clinics can improve spectacle access for children with RE. The survey further reveals that most guardians are satisfied with current inventories. Based on their positive views of the

convenience and quality of current service, guardians indicated they would return in the future and would recommend the clinics to others. Design, material, and quality of frames are key factors influencing guardians in this setting when purchasing spectacles for their children, although female guardians and those with higher income prioritise frame quality. This study can inform the planning and implementation of a planned novel cross-subsidisation scheme in Cross River State.

## Supporting information

**S1 File. Survey questionnaire to elicit factors affecting guardians' decision making on clinic-based purchase of children's spectacles in Nigeria.**
(DOCX)

## Acknowledgments

We want to thank Drs Lovelyn George, Ogechi Nneji, Emmanuel Kalu, Priscilla Chukwu, C. Emedike, Kenneth Azubuike, Emmanuel Iwong, Emmanuel Odor and Ms Okwoalice Ita for collecting the data; and all the guardians who participated in the study.

## Author Contributions

**Conceptualization:** Ving Fai Chan, Ciaran O'Neill, Nathan Congdon, Lynne Lohfeld, Anne Effiom Ebri.

**Data curation:** Ving Fai Chan, Ai Chee Yong.

**Formal analysis:** Ving Fai Chan, Ai Chee Yong, Nathan Congdon.

**Funding acquisition:** Ving Fai Chan, Ciaran O'Neill, Nathan Congdon, Anne Effiom Ebri.

**Investigation:** Ving Fai Chan, Nathan Congdon, Lynne Lohfeld, Anne Effiom Ebri.

**Methodology:** Ving Fai Chan, Ciaran O'Neill, Nathan Congdon, Lynne Lohfeld, Anne Effiom Ebri.

**Project administration:** Ving Fai Chan, Christine Graham, Anne Effiom Ebri.

**Supervision:** Christine Graham, Anne Effiom Ebri.

**Validation:** Ving Fai Chan, Christine Graham, Anne Effiom Ebri.

**Visualization:** Ving Fai Chan, Ai Chee Yong.

**Writing – original draft:** Ving Fai Chan, Ai Chee Yong.

**Writing – review & editing:** Ving Fai Chan, Ai Chee Yong, Ciaran O'Neill, Christine Graham, Nathan Congdon, Lynne Lohfeld, Tai Stephan, Anne Effiom Ebri.

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
