## [Decision Letter · Decision Letter 0]

26 May 2021

PONE-D-21-12910

Factors Affecting Guardians' Decision Making on Clinic-based Purchase of Children's Spectacles in Nigeria

PLOS ONE

Dear Dr. Chan,

Thank you for submitting your manuscript to PLOS ONE. After careful consideration, we feel that it has merit but does not fully meet PLOS ONE’s publication criteria as it currently stands. Therefore, we invite you to submit a revised version of the manuscript that addresses the points raised during the review process.

The study has some flaws in the methodology such as the sampling technique and other confounding variables that were not explored. This should be addressed by the authors.

We look forward to receiving your revised manuscript.

Kind regards,

Ahmed Awadein, MD, Ph.D, FRCS

Academic Editor

PLOS ONE

Journal Requirements:

Furthermore, please provide additional details regarding the steps taken to validate the questionnaire.

Finally, in your Methods section, please provide a justification for the sample size used in your study, including any relevant power calculations (if applicable).

"VFC is a Trustee of Vision Aid Overseas, a non-governmental organisation involved in delivering refractive services in LMICs, including Africa. NC is the Director of Research for Orbis International, an organisation involved in delivering eyecare, including children's refractive services, in Africa and other settings."

Reviewers' comments:

Reviewer's Responses to Questions

**Comments to the Author**

1. Is the manuscript technically sound, and do the data support the conclusions?

Reviewer #1: Yes

Reviewer #2: No

2. Has the statistical analysis been performed appropriately and rigorously? 

Reviewer #1: Yes

Reviewer #2: No

3. Have the authors made all data underlying the findings in their manuscript fully available?

Reviewer #1: No

Reviewer #2: Yes

4. Is the manuscript presented in an intelligible fashion and written in standard English?

Reviewer #1: Yes

Reviewer #2: Yes

5. Review Comments to the Author

Reviewer #1: This is a very nicely done study demonstrating the perceptions of guardians regarding their children's eyewear in one part of Nigeria. Overall, the manuscript is well laid out and presented, furthering earlier similar studies conducted in China and South Africa.

Minor comments and suggested corrections involve mainly accurate reporting of some of the results (please note that the PDF file that I am referencing did not have line numbers included):

Abstract, Results (p. 7): "n=92" is confusing here; reword as "92 females", "67.2% female [n=92]" or similar, as you have done in the main text (p. 12).

Results (p. 12): please clarify how you determined the cut-off monthly income as USD263. Table 1 uses a criterion value of USD240. Did you ask each guardian what their actual income is? (I could not find a link to the underlying data.)

Results (p. 13): please check the values reported in the text against what you report in Table 2; 58.5% should be 58.8% and 12.4% should be 14.2%, if I read the table correctly.

Reviewer #2: A good question was posed by the authors on the decision criteria of guardians for choosing a spectacle for their child.

But methods of the paper does not justify the results based on which further conclusions can be made. An added objective of a plan for cross- subsidy further weakens the conclusions of the study.

Authors aim to see the factors influencing purchase decision of spectacles, however these were posed as closed ended questions with pre determined limited options. A broader perspective of the guardians was limited, which is the main expected outcome. There could be many other factors that influences the decision of the guardians apart from the ones explored, adding to the limitation of the study. Improvement in vision alone could have been the key reason.

Details on the available choices and range of the brands, materials, types and cost of spectacles would give a holistic picture based on which the guardian's decision relied on.

Sampling of the study- not completely representative and less sample size: Samples selected from the semi urban and urban were those who visited the hospital seeking care and hence in the mindset of availing care. It should have had samples from non-seekers too as they intend to see whether this would work as a cross- subsidy scheme for their region.

Very minimal representation of sample from each of the arms, hence extrapolation is difficult. Urban representation is far less

Questionnaire that was used could be shared

If the ultimate aim is to see whether the cross subsidy could be implemented, (based on the factors) many more factors apart from the details on the optical material, style and quality considerations are required.

6. PLOS authors have the option to publish the peer review history of their article (what does this mean?). If published, this will include your full peer review and any attached files.

Reviewer #1: No

Reviewer #2: No

---

## [Author Response · Author response to Decision Letter 0]

23 Jun 2021

Responses to reviewers 

Comments to the Author:

Response: Thanks for highlighting this. We have amended the files in accordance to the style templates. 

Furthermore, please provide additional details regarding the steps taken to validate the questionnaire.

Finally, in your Methods section, please provide a justification for the sample size used in your study, including any relevant power calculations (if applicable).

Response: The questionnaire is included as Supporting information. The questionnaire was first reviewed by a panel of experts from the field of public health, ophthalmology, anthropology and optometry, and then piloted with ten local community members to ensure its appropriateness, relevance, comprehensiveness. Minor changes were made to include closed-ended questions on why guardians chose to purchase particular spectacles for their children in eye clinics and their perceptions of the quality and design of children's current spectacle (5-point Likert-scale, 1 = Very bad and 5 = excellent) in the questionnaire.

Power calculation was not conducted as this is a non-interventional study and observational in nature with no hypothesis testing. We employed a convenient finite sampling where all guardians visiting participating eye clinics from 1 August to 31 October 2019 were interviewed if their children had significant REs (myopia ≤-0.50D, hyperopia ≥1.50D, astigmatism >0.75D) and received spectacles. This yielded 137 guardians. 

These information were included in Methods and material.

"VFC is a Trustee of Vision Aid Overseas, a non-governmental organisation involved in delivering refractive services in LMICs, including Africa. NC is the Director of Research for Orbis International, an organisation involved in delivering eyecare, including children's refractive services, in Africa and other settings."

Response: The conflict of interest stated in the manuscript does not alter our adherence to PLOS ONE policies on sharing data and materials. We have included this in both the manuscript and covering letter.

Reviewer #1: This is a very nicely done study demonstrating the perceptions of guardians regarding their children's eyewear in one part of Nigeria. Overall, the manuscript is well laid out and presented, furthering earlier similar studies conducted in China and South Africa. Minor comments and suggested corrections involve mainly accurate reporting of some of the results (please note that the PDF file that I am referencing did not have line numbers included):

Comment 1:

Abstract, Results (p. 7): "n=92" is confusing here; reword as "92 females", "67.2% female [n=92]" or similar, as you have done in the main text (p. 12).

Response: Thank you for highlighting this. We have amended this as suggested.

Comment 2:

Results (p. 12): please clarify how you determined the cut-off monthly income as USD263. Table 1 uses a criterion value of USD240. Did you ask each guardian what their actual income is? (I could not find a link to the underlying data.)

Response: Thanks for highlighting the error. It should be USD240 and not USD263 as per Table1. Each guardian was asked an open-ended question on their actual monthly income. After discussing a suitable cut-off for low vs high income with local partners, we then regrouped them into two categories: ≤ 100,000 Nigerian Naira (USD240), and > 100,000 Nigerian Naira. Because the data were not normally distributed, we also provided the median and interquartile range.

The following statement “Each guardian was asked an open-ended question on their actual monthly income. After discussing a suitable cut-off for low vs high income with local partners, we then regrouped them into two categories: ≤ 100,000 Nigerian Naira (USD240), and > 100,000 Nigerian Naira.” is included in Methods.

Comment 3:

Results (p. 13): please check the values reported in the text against what you report in Table 2; 58.5% should be 58.8% and 12.4% should be 14.2%, if I read the table correctly.

Response: Thank you for spotting this. We made the amendment accordingly.

Reviewer #2: A good question was posed by the authors on the decision criteria of guardians for choosing a spectacle for their child. But methods of the paper does not justify the results based on which further conclusions can be made. An added objective of a plan for cross- subsidy further weakens the conclusions of the study.

Comment 4:

Authors aim to see the factors influencing purchase decision of spectacles, however these were posed as closed ended questions with pre determined limited options. A broader perspective of the guardians was limited, which is the main expected outcome. There could be many other factors that influences the decision of the guardians apart from the ones explored, adding to the limitation of the study. Improvement in vision alone could have been the key reason.

Details on the available choices and range of the brands, materials, types and cost of spectacles would give a holistic picture based on which the guardian's decision relied on.

Response: Thank you for highlighting this. To cover a broader perspective beyond frame choices, we also included an open-ended question asking guardians to state if any other factors influenced their decisions when purchasing spectacles. We grouped the resultant factors into five sets: It was convenient, I like the spectacle frames, The service was good, I didn’t know I could get it elsewhere, or Others (which included free treatment, because this is a hospital, referred, general health check, government establishment offers a better price, and the staff is friendly). These are included in Table 2 (highlighted in red). 

Comment 5:

Sampling of the study - not completely representative and less sample size: Samples selected from the semi urban and urban were those who visited the hospital seeking care and hence in the mindset of availing care. It should have had samples from non-seekers too as they intend to see whether this would work as a cross- subsidy scheme for their region.

Very minimal representation of sample from each of the arms, hence extrapolation is difficult. Urban representation is far less.

Response: Thank you for highlighting this limitation with regards to the smaller sample size and convenience sampling method used in this study. Furthermore, even though the clinics are situated in semi-urban and urban areas, they serve all strata of populations (rural, semi-urban and urban). We have included the statement in the Introduction, “To assist with the design of the children’s spectacles cross-subsidisation strategy that will be implemented in CRS, a four-part study was conducted. It collected collect data on child eye morbidities prevalence; barriers to follow-up eye examinations and spectacle uptake; parents’ willingness to pay for their children’s spectacles; and the current study which aims to understand the key factors influencing guardians' decisions when purchasing spectacles for their children in semi-urban (Ikom, Ugep and Ogoja) and urban (Calabar) areas. These four clinics serve the rural, semi-urban and urban population in CRS. ” 

Lohfeld et al. also explored the reasons for those who did not seek treatment and the different factors that will encourage them to seek treatment. (https://medrxiv.org/cgi/content/short/2021.06.08.21258336v1). 

We have reworded this limitation to the following: “Thirdly, we recognized that some parents may not have brought their children for treatment at one of the local hospitals and so their views may have been unrepresented. We addressed this concern in another study in which we interviewed guardians who did not bring their children for a follow-up eye examination at a hospital eye clinic. The adults were asked about reasons for not seeking additional treatment and what would encourage them to do so (https://medrxiv.org/cgi/content/short/2021.06.08.21258336v1).” 

Comment 6:

Questionnaire that was used could be shared.

Response: Thank you for suggesting this. We will upload the questionnaire as supplementary material.

Comment 7:

If the ultimate aim is to see whether the cross subsidy could be implemented, (based on the factors) many more factors apart from the details on the optical material, style and quality considerations are required.

Response: We agreed on this valid point. This study is part of a larger study conducted in Nigeria which seeks to understand guardians’ selection criteria and willingness-to-pay for public sector children’s spectacles in assisting in eye care programme planning and determining the correct pricing for a cross-subsidization scheme that will be (rather than could be) implemented in Nigeria. Hence, besides factors influencing guardians’ decision making in purchasing children’s spectacles, our other research teams also looked at factors affecting their willingness to pay for their children’s spectacles; mapping the demographic profile and disease prevalence of children attending follow-up eye examinations (demand); and understanding barriers to the child eye services uptake. The barrier study has been submitted (https://medrxiv.org/cgi/content/short/2021.06.08.21258336v1) and another two papers will be submitted to peer-reviewed journals.

To avoid this confusion, we reworded our statement in the Introduction to the following: “To assist with the design of the children’s spectacles cross-subsidisation strategy that will be implemented in CRS, a four-part study was conducted. It collected collect data on child eye morbidities prevalence; barriers to follow-up eye examinations and spectacle uptake; parents’ willingness to pay for their children’s spectacles; and the current study which aims to understand the key factors influencing guardians' decisions when purchasing spectacles for their children in semi-urban (Ikom, Ugep and Ogoja) and urban (Calabar) areas. These four clinics serve the rural, semi-urban and urban population in CRS.

---

## [Editor Report · Decision Letter 1]

29 Jun 2021

Factors Affecting Guardians' Decision Making on Clinic-based Purchase of Children's Spectacles in Nigeria

PONE-D-21-12910R1

Dear Dr. Chan,

We’re pleased to inform you that your manuscript has been judged scientifically suitable for publication and will be formally accepted for publication once it meets all outstanding technical requirements.

Kind regards,

Ahmed Awadein, MD, Ph.D, FRCS

Academic Editor

PLOS ONE
---

## [Editor Report · Acceptance letter]

1 Jul 2021

PONE-D-21-12910R1 

Factors Affecting Guardians' Decision Making on Clinic-based Purchase of Children's Spectacles in Nigeria 

Dear Dr. Chan:

I'm pleased to inform you that your manuscript has been deemed suitable for publication in PLOS ONE. Congratulations! Your manuscript is now with our production department. 

Kind regards, 

on behalf of

Dr. Ahmed Awadein 

Academic Editor

PLOS ONE